# Contrasting and prioritizing dimensions in ethnic teacher education: A convergent analysis with LDA and fsQCA

**Qimeng Wu**[1]*, **Qiyun Wu**[2]

**1** School of History, Qinghai Normal University, Xining, China, **2** School of Education, Qinghai Normal University, Xining, China

\* 2360789363@qq.com

## Abstract

Previous studies have rarely examined ethnic teacher education from a configurational perspective. This study addresses that gap.This study analyzes 250 peer-reviewed articles from CNKI's PKU/CSSCI journals (2015–2025), identified using 'ethnic teacher education/education/teachers' keywords, refined from 2,672 initial records through manual screening.This study employs a mixed-methods approach to investigate ethnic teacher education in China, combining computational text analysis with configurational methods. This study employs ROST Content Mining 6.0 for semantic network analysis and CiteSpace 6.3 for knowledge mapping. The analysis reveals five core conceptual clusters (development, teachers, ethnicity, education, and region) and five research hotspots, with 'ethnic regions' emerging as the dominant focus. Through LDA topic modeling, five thematic dimensions were extracted, ranging from the status of ethnic education to cultural integration in teacher training. These qualitative insights were subsequently quantified and analyzed via fsQCA, yielding six distinct optimization pathways. The findings suggest a dual trajectory for future research: while emphasizing geo-cultural specificity in teacher training program design, the study also identifies universal principles applicable across ethnic education contexts. Accordingly, five policy recommendations are proposed, including diversifying development channels, enhancing cultural responsiveness in pedagogy, and facilitating inter-regional knowledge exchange among ethnic communities. This study introduces the novel integration of LDA topic modeling and fsQCA in educational research. While LDA uncovers latent themes and fsQCA examines causal complexity, their combined application enables simultaneous discovery and validation of configurations a previously unexplored approach in ethnic teacher education.These findings make dual contributions: theoretically, by advancing a novel conceptual framework; and practically, by yielding actionable policy implications for ethnic teacher education development.

**Data availability statement:** All relevant data are within the paper and its Supporting Information files.

**Funding:** The author(s) received no specific funding for this work.

**Competing interests:** The authors have declared that no competing interests exist.

## 1. Introduction

As a vital component of China's basic education system, ethnic education serves dual functions in facilitating socioeconomic progress and preserving cultural heritage in ethnic regions [1]. The quality of ethnic teacher education proves particularly pivotal, directly determining educational standards and future development trends in these areas. The Chinese government has demonstrated sustained commitment through progressive policy measures, notably the 2018 Guidelines on Deepening Teacher Team Reform in the New Era, which addressed critical imbalances between infrastructure and human resource investment while emphasizing teacher capacity building [2]. This policy orientation was further refined in 2022 through the Ministry of Education's initiative to develop high-quality educators and enhance teacher education systems [3].

Nevertheless, ethnic teacher education faces compounded challenges stemming from three distinctive regional characteristics: unique geographical conditions, diverse cultural environments, and relatively underdeveloped economic foundations [4]. These factors collectively contribute to multiple constraints, including resource scarcity in remote areas, limited professional development opportunities, and persistent mismatches between current teacher training systems and the complex realities of multicultural classrooms – particularly evident in curricular design and program objectives.

Given the fragmented understanding of ethnic teacher education in China, this study examines: (1) the core thematic dimensions of ethnic teacher education, and (2) how their configurations influence educational effectiveness. To address these challenges, this study adopts a methodological approach combining textual analysis and fuzzy-set Qualitative Comparative Analysis (fsQCA) to: (1) identify critical research priorities, (2) analyze current developmental trends, and (3) establish empirically-grounded optimization pathways. This study innovatively integrates LDA and fsQCA to overcome regression analysis' limitations in studying ethnic teacher education. Unlike regression's linear assumptions [5] (Gelman & Hill, 2007), our approach captures system complexity and equifinality [6] (Fiss, 2011) through two phases: LDA [7] (Blei et al., 2003) extracts cultural themes from qualitative data, then fsQCA [8] (Ragin, 2009) analyzes their configurations. Key advantages include: (1) small-N robustness with contextual sensitivity [9] (Berg-Schlosser et al., 2009), (2) identification of causal asymmetries [10] (Misangyi et al., 2017), and (3) multiple success pathways for diverse contexts. This advances understanding of ethnic education's complex causality beyond variable-centered methods. The research outcomes aim to provide both theoretical frameworks and practical guidance for reforming China's teacher education models, ultimately contributing to the cultivation of high-quality educators tailored to the specific needs of ethnic regions. These findings hold significant implications for achieving equitable and sustainable educational development across China's diverse regional contexts.

Recent years have witnessed growing scholarly attention to ethnic teacher education in China, reflecting its crucial role in the nation's educational landscape [11]. While existing studies have predominantly focused on analyzing professional skill development

among teachers in ethnic minority regions, this narrow perspective fails to capture the comprehensive evolution of teacher education as an integrated system [12]. The ongoing modernization of China's education sector, coupled with the implementation of national unity education policies, has fundamentally transformed ethnic teacher education from a skill-based training model to a multidimensional educational practice. This paradigm shift now encompasses language and cultural education, social integration, and the cultivation of professional competencies, marking a significant advancement in the field.The unique professional demands placed on ethnic teachers necessitate a delicate equilibrium between delivering standardized national curricula and preserving rich cultural traditions. This dual responsibility creates distinct challenges in teacher development that require specialized attention. Current research methodologies, however, remain largely retrospective, concentrating on evaluating existing educational practices rather than anticipating future trends.The present study addresses this critical gap by employing innovative analytical approaches, including fuzzy-set qualitative comparative analysis (fsQCA), to identify key developmental pathways and predict emerging directions in ethnic teacher education. Our research particularly emphasizes the strategic integration of intelligent educational technologies with culturally responsive pedagogy, proposing a forward-looking framework for teacher professional development.These findings carry significant implications for educational policy and practice, offering both theoretical insights and practical strategies to enhance teacher education systems in ethnic regions. By bridging the divide between technological advancement and cultural preservation, this study contributes to the creation of more inclusive and effective educational environments that support both academic excellence and cultural continuity in China's diverse ethnic communities.

The theory of educational ecology is an interdisciplinary application. Its theoretical foundation stems from the principles and methods of ecology that are applied to the field of education. This theory originated when Lawrence A. Tyack [13], the dean of the Teachers College at Columbia University in the United States, creatively put forward the theory of "educational ecology." He defined education as "a deliberate, systematic, and continuous effort to evoke knowledge, attitudes, values, skills, and emotions." At the same time, in his theory, he pointed out that the theoretical basis of educational ecology is the "interaction theory," emphasizing that various educational institutions are interconnected and influence each other, as well as having a mutual impact with the entire society. In the system of educational ecology theory, it also adheres to the principles of ecological balance, ecological adaptation, and ecological interaction. All elements within the ecological theory need to maintain a dynamic balance. The theory of educational ecology also focuses on issues such as the distribution and composition of the population and interpersonal relationships. It is committed to establishing a reasonable ecological environment both inside and outside schools to improve teaching efficiency and promote the all-round development of individuals. In the education of ethnic minority teachers, the education they receive influences their behaviors, and the education of ethnic minority teachers, guided by the theory of educational ecology, affects the outcomes of ethnic minority education. In this study, the development trends of ethnic minority education are predicted by examining the research hotspots in the education of ethnic minority teachers, and then developmental suggestions for the education of ethnic minority teachers are put forward. The Teaching Behavior-Learning Outcome Model:Mao, G., & Liu, Q. T. (2020) [14] proposed the relationship between teaching behaviors and learning effects, providing a fundamental theoretical perspective for understanding this model. Moreover, some scholars, through empirical research, have further verified the correlations between various elements of teaching behaviors and learning outcomes, providing data support for the scientific nature of the model. The Teaching Behavior-Learning Outcome Model elucidates the inherent connection between teachers' behaviors and students' learning effects. This model posits that teachers' teaching behaviors are one of the crucial factors influencing students' learning outcomes. In ethnic minority education, the quality of ethnic minority education can be enhanced by focusing on the research priorities of the education of ethnic minority teachers.

## 2. Methods

### 2.1. Research object and data source

In this article, the research objects mainly originate from the CNKI database. The retrieval fields selected are the subject, article title, and the abstract with key points. The search terms chosen are "ethnic minority teacher education", "ethnic

 

minority education", or "ethnic minority teachers". This study exclusively analyzed peer-reviewed journal articles focusing on teacher training programs within Mainland China.The retrieval scope is limited to the catalogs of Peking University Core Journals and Nanjing University Core Journals. The time span for the retrieval is set from 2015 to April 2025. The total retrieval results encompass 2,672 pieces of literature, among which there are 399 core literature articles in total. Article selection followed PRISMA guidelines for duplicate removal, with two researchers independently screening studies using the CASP Qualitative Checklist. Discrepancies were resolved through consensus. Through manual reading of the literature abstracts, data that do not meet the requirements are removed. After integrating the obtained results, duplicate literature is eliminated. Finally, 250 valid pieces of literature are obtained.

The literature abstracts are imported into the ROST Content Mining 6.0 text analysis tool for high-frequency word analysis and semantic network analysis. The data are then imported into Citespace 6.3 to generate a visualized knowledge graph for sorting out the research focuses. The stop word removal and word segmentation operations are carried out by using the stop word library of Harbin Institute of Technology that comes with the Jieba package in Python. Subsequently, the LDA (Latent Dirichlet Allocation) topic clustering operation is performed. After clustering, fuzzy set values are assigned to the topic dimensions according to the 250 pieces of research literature. The fsQCA (fuzzy-set Qualitative Comparative Analysis) is applied for configurational research. On this basis, corresponding optimized paths for the development of ethnic minority teacher education are proposed. The specific flowchart is shown in Fig 1.

## 2.2. Measures

**2.2.1. Text analysis employing ROST content mining 6.0.** This research made use of the ROST Content Mining 6.0 text analysis platform, which incorporates three pivotal functional modules: Text Preprocessing: This module features custom vocabulary setup, application of filter lists, and Chinese word segmentation. These functions are crucial for standardizing the raw text data, laying a solid foundation for subsequent analyses. Quantitative Analysis: It has the capacity to conduct word frequency statistics and social network analysis. By doing so, it can effectively identify prominent lexical patterns and relational structures within the text, which are essential for understanding the text's surface – level characteristics. Deep Mining: Tools for constructing semantic networks and recognizing emotional tendencies are included

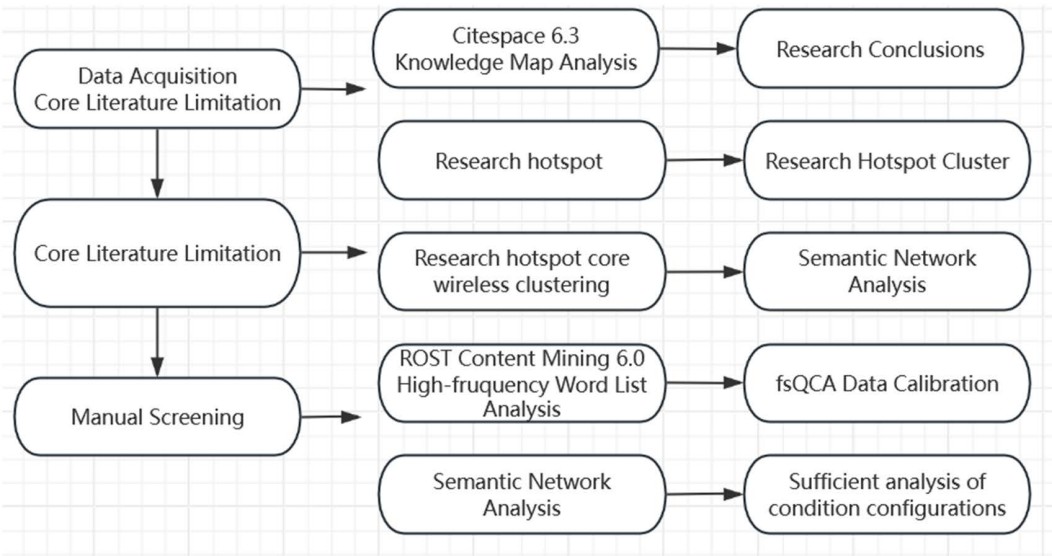

**Fig 1. Technical flow chart.**

in this module. They are designed to decipher the implicit semantic and affective dimensions hidden in the text, thus providing a more in – depth understanding [15].

The analytical process consisted of three main stages: Data Collection: Raw text data, such as online travelogues, were gathered from targeted sources. These data served as the primary material for the entire research. Feature Extraction: Through iterative word segmentation and frequency statistics, feature word lists were generated. During this process, terms with high salience scores were given priority, as they are more likely to carry important information. Dimension Parsing: Semantic network visualizations and high – frequency word distributions were integrated to operationalize three constructs related to the destination image: Cognitive Dimensions: Measured by the attention levels to elements. For instance, the frequency and centrality of destination attributes in semantic networks can reflect how tourists cognitively perceive the destination. Emotional Dimensions: Inferred from the distribution of emotional lexicons. The patterns of positive/negative sentiment words and their co – occurrence can reveal the emotional stances of tourists. Overall Perceptions: Synthesized through semantic associations. The strength of co – occurrence between cognitive and emotional terms helps to form an overall picture of tourists' perceptions.

**2.2.2. Topic modeling using Latent Dirichlet Allocation (LDA).** The Latent Dirichlet Allocation (LDA) model [16], a probabilistic topic – modeling framework, was adopted to unearth latent semantic structures in user – generated content. As a typical bag – of – words model, LDA operates on the premise that documents are composed of unordered word combinations. It employs unsupervised learning to deduce latent topics from text corpora. The model uses a three – tier Bayesian structure – document – topic – word – to construct two probability matrices: A "document - topic" matrix that represents the distribution of topics within each document, indicating which topics are more prominent in a particular document. A "topic - word" matrix that encodes the probability of words belonging to each topic, showing the relationship between words and topics.

By analyzing word co-occurrence patterns, LDA can identify thematic clusters without the need for labeled training data. This characteristic makes it highly suitable for semantic feature extraction, text clustering, and trend detection in natural language processing tasks [17] Leveraging its proven effectiveness in analyzing user – generated content, such as online reviews, this study applied LDA to mine recurring themes in tourism reviews of Tibetan culture in Qinghai Province. The aim was to identify the core concerns of tourists and the discursive patterns in these reviews.

**2.2.3. Configurational analysis with fuzzy - Set qualitative comparative analysis (fsQCA).** Fuzzy – set qualitative comparative analysis [18] was chosen as the main analytical method because of its advantages in small – to – medium sample research. In such research, traditional regression – based techniques often face limitations. Unlike variable – centric approaches, fsQCA takes a holistic, set – theoretic perspective to explore how configurations of conditional variables result in specific outcomes. Its key strengths are as follows:

Equifinality Analysis: It can identify multiple variable combinations that lead to equivalent outcomes [19]. This is particularly useful in educational research, where educational phenomena are complex and a single causal explanation is often inadequate [20].

Contextual Sensitivity: It reveals how the same variable may play different causal roles across different configurations, enabling a more nuanced understanding of nonlinear relationships.

The fsQCA framework includes three variants: crisp – set QCA (csQCA), fuzzy – set QCA (fsQCA), and multi – value QCA (mvQCA). This study used fsQCA for two crucial reasons:

It relaxes the strict binary (0/1) variable coding of csQCA by allowing continuous membership scores in the [0,1] interval. This is more in line with the graded nature of educational variables, such as teacher competence and institutional support. It facilitates the identification of complex causal pathways, including both convergent (multiple configurations leading to the same outcome) and divergent (the same configuration leading to different outcomes) relationships. As a result, it enhances the interpretive power of small sample analyses.

The analytical steps were as follows:

Variable Calibration: Raw data were converted into fuzzy – set membership scores using theoretically – grounded calibration procedures. This step was essential for preparing the data for fsQCA analysis. Truth Table Construction: Configurations of conditional variables and their associated outcome membership scores were formalized. The truth table served as the basis for subsequent analyses [21]. Solution Term Extraction: Boolean minimization was applied to identify parsimonious and intermediate solutions that represent sufficient conditions for the target outcome.

This approach enabled a systematic exploration of how configurations of teacher education inputs, such as curriculum design, cultural competence training, and institutional resources, contribute to outcomes in ethnic minority teacher education. It also provided actionable insights for educational policies and practices.

## 3. Results

### 3.1. Research focus and projections in ethnic minority teacher education

Extraction of high-frequency terms in ethnic minority teacher education research.

This study utilized ROST Content Mining 6 software to conduct word frequency analysis on texts retrieved from CNKI. By applying the Harbin Institute of Technology stop word list and a custom lexicon to filter out nonsensical vocabulary, 30 high-frequency terms related to ethnic minority teacher education were extracted (see Table 1). For analytical consistency, semantic synonyms were merged: our country and country were unified as country; foundation and basic education were consolidated into foundation; and promote and enhance were standardized as enhance.

Part-of-Speech Analysis revealed nouns as the dominant lexical category, including key proper nouns specific to the field such as ethnic group, bilingualism, and ethnic minorities. The high-frequency term list also featured a significant presence of positive action verbs, with terms like development,enhancement, cultivation and construction appearing prominently—indicating an overall proactive orientation in research trends. Lexical Composition was as follows: Nouns: 14 terms (47% of the total), highlighting substantive concepts central to ethnic minority teacher education; Verbs: 14 terms (47% of the total), emphasizing actionable focus areas like capacity building and systemic improvement; Adjectives: 2 terms (6% of the total), reflecting qualitative dimensions of the research landscape.

Analyzing these high-frequency terms enables precise identification of core research directions and priorities in ethnic minority teacher education. This lexical categorization not only enhances the interpretability of research trends but also

**Table 1. High-frequency word list of research focus in ethnic minority teacher education.**

| Serial Number | Word | Frequency | Part of Speech | Serial Number | Word | Frequency | Part of Speech |
|---|---|---|---|---|---|---|---|
| 1 | Teacher | 770 | Noun | 16 | Research | 110 | Verb |
| 2 | Education | 764 | Verb | 17 | Foundation | 107 | Adjective |
| 3 | Ethnic Group | 590 | Noun | 18 | Quality | 106 | Noun |
| 4 | Region | 397 | Noun | 19 | Team | 101 | Noun |
| 5 | Development | 337 | Verb | 20 | Ability | 101 | Verb |
| 6 | Culture | 225 | Noun | 21 | Policy | 99 | Noun |
| 7 | Teaching | 196 | Verb | 22 | Training | 98 | Verb |
| 8 | Ethnic Minorities | 165 | Noun | 23 | School | 89 | Noun |
| 9 | Enhancement | 157 | Verb | 24 | Teaching Staff | 88 | Noun |
| 10 | Cultivation | 149 | Verb | 25 | Music | 85 | Verb |
| 11 | Country | 139 | Noun | 26 | Student | 84 | Noun |
| 12 | Problem | 124 | Noun | 27 | Practice | 77 | Verb |
| 13 | Construction | 116 | Verb | 28 | Promotion | 74 | Verb |
| 14 | Curriculum | 116 | Noun | 29 | Improvement | 72 | Verb |
| 15 | Bilingual | 112 | Adjective | 30 | Existence | 70 | Verb |

provides a robust foundation for subsequent investigations, ensuring alignment with the field's developmental needs and policy objectives.

### 3.2. Visual analysis of ethnic teacher education research

Guided by the holistic educational theory model, this study conducts a semantic network analysis by importing raw data into ROST Content Mining 6 software to examine ethnic teacher education comprehensively. Through analyzing online texts, it is found that the construction of the overall research framework in ethnic teacher education lacks support from theoretical models. While high-frequency word lists can reflect research hotspots, they fail to intuitively illustrate the interconnections between terms or the composition of their connotations. A semantic network analysis graph consists of two components: nodes, which represent entities, states, emotions, etc., and arcs, which denote the semantic relationships between nodes. By applying the semantic network analysis function, a visualized graph was generated to depict these relationships, as shown in Fig 2.

It was found that this study identifies five core word clusters radiating outward to secondary clusters in a star-like pattern. The core terms are "development," "teachers," "ethnicity," "education," and "region," as illustrated in Fig 2. Surrounding these core clusters are secondary word clusters that complement and expand the core concepts. Since the educational target of ethnic teacher education is teachers, the term "teachers" serves as a central node in the semantic network, radiating to secondary terms such as "competency," "curriculum," "training," "teaching staff," "improvement," and "instruction." The term "ethnicity," representing the unique characteristic of ethnic teacher education, is linked to secondary terms like "schools," "foundation," and "quality," reflecting the special requirements of teacher education in regions with distinct natural and cultural environments. The core term "education" connects to secondary terms such as "practice," "ethnic minorities," "promotion," and "culture," while "region"—grounded in the socioeconomic and geographical distinctiveness of ethnic areas—associates with "teaching resources," "ethnic minorities," and "improvement." The core term "development" radiates to terms like "teaching," "culture," and "research," indicating its role in driving educational progress.

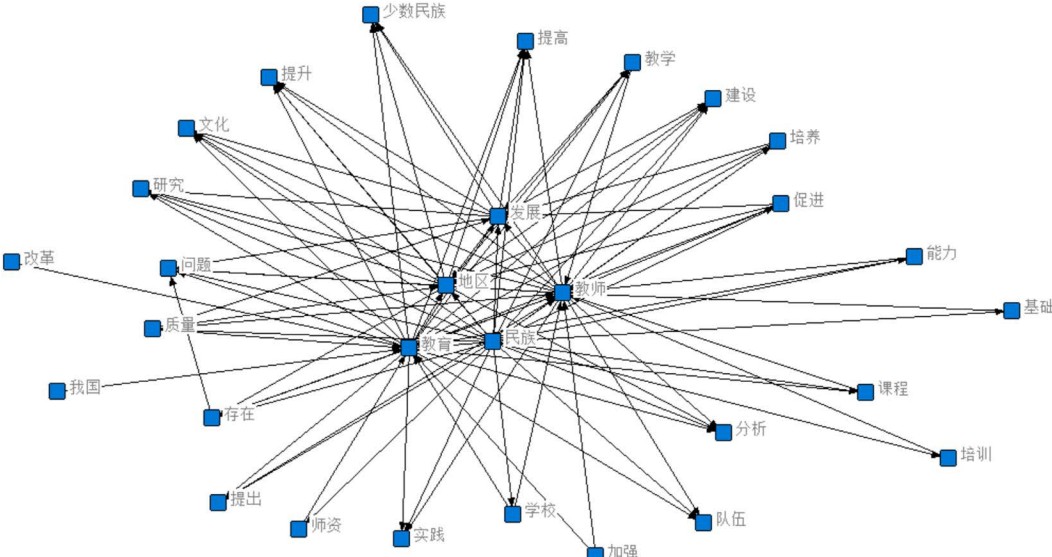

**Fig 2. Semantic network analysis diagram of the research focus on ethnic teacher education.** 文化/ Culture; 研究/ Research; 问题/ Issues (or Problems); 质量/ Quality; 我国/ Our Country; 存在/ Existence; 提出/ Proposal; 师资/ Teaching Staff (or Faculty); 实践/ Practice; 提高/ Enhancement (or Improvement); 教学/ Teaching; 建设/ Development (or Construction); 培养/ Cultivation; 促进/ Promotion; 能力/ Capability (or Ability); 基础/ Foundation; 课程/ Curriculum (or Courses); 分析/ Analysis; 培训/ Training; 队伍/ Team.

The scattered distribution of secondary terms in the graph suggests that research focuses in ethnic teacher education are broad but lack systematic integration with core concepts.

After exporting selected literature from CNKI and importing it into Citespace 6.3, 250 valid documents were standardized and analyzed using author, institution, and keyword nodes to generate visual knowledge maps through node-clustering. To identify research hotspots and trends in ethnic teacher education, a keyword co-occurrence map was created by selecting "Key words" as the node type (Fig 3). The analysis revealed six major research clusters: "ethnic regions," "ethnic education," "teacher education," "rural education," "teaching staff," and "teacher education" (note: potential repetition, retained as per original). "Ethnic regions" emerged as a central cluster, serving as both the contextual foundation and research focus of ethnic teacher education. Interactions between "ethnic regions" and "rural teachers" or "basic education" indicate a primary focus on foundational education in these areas. The intersection of "ethnic regions" and "teacher education" highlights the latter's critical role in the educational system. However, the "teaching staff" cluster showed limited connectivity with other hotspots, forming an isolated research focus. This lack of interaction among key clusters contributes to the fragmented nature of ethnic teacher education research, undermining its overall systematic impact.

In Citespace6.3, a co-occurrence network analysis of research institutions visually maps the key entities in China's ethnic education research field. As shown in Fig 4, nodes represent individual institutions, with node size proportional to their cooperation frequency and connection thickness indicating the intensity of inter-institutional collaboration. The analysis reveals four primary institutional clusters:

North China Cluster: Centered on Beijing Normal University and Minzu University of China, representing the core research hubs in northern China; Southwest Cluster: Headquartered at Southwest University, dominating academic activities in the southwestern region; Northwest Cluster: Led by Shihezi University, serving as the regional focal point for northwest China; Northeast Cluster: Primarily anchored by Harbin Normal University, representing research strengths in northeastern China.

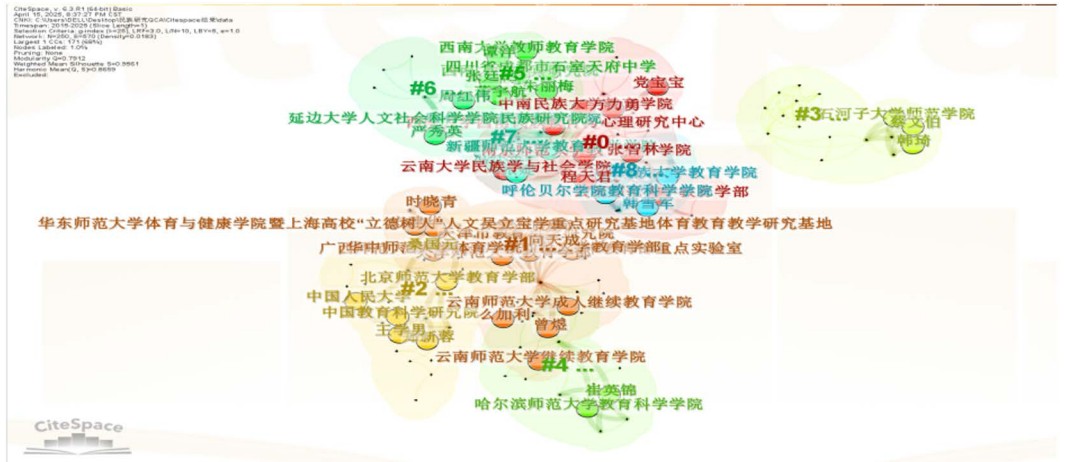

**Fig 3. Research hotspot map of ethnic teacher education.** 华东师范大学体育与健康学院暨上海高校"立德树人"人文实验室学生重点研究基地体育教育教学研究基地 | School of Sports and Health, East China Normal University & Shanghai Key Research Base for "Moral Education" in Sports Pedagogy;广西梅州师范学校附属大学附属天成九院 | Tiancheng Ninth Affiliated College, Meizhou Normal School, Guangxi;北京师范大学教育学院 | School of Education, Beijing Normal University;中国人民大学 云南师范大学成人继续教育学院 | Renmin University of China & Yunnan Normal University College of Adult and Continuing Education;中国教育科学研究院 | China National Institute of Education Sciences (Jia Li, Zeng Yu); 云南师范大学继续教育学院 | College of Continuing Education, Yunnan Normal University; 哈尔滨师范大学教育科学学院 | School of Educational Science, Harbin Normal University.

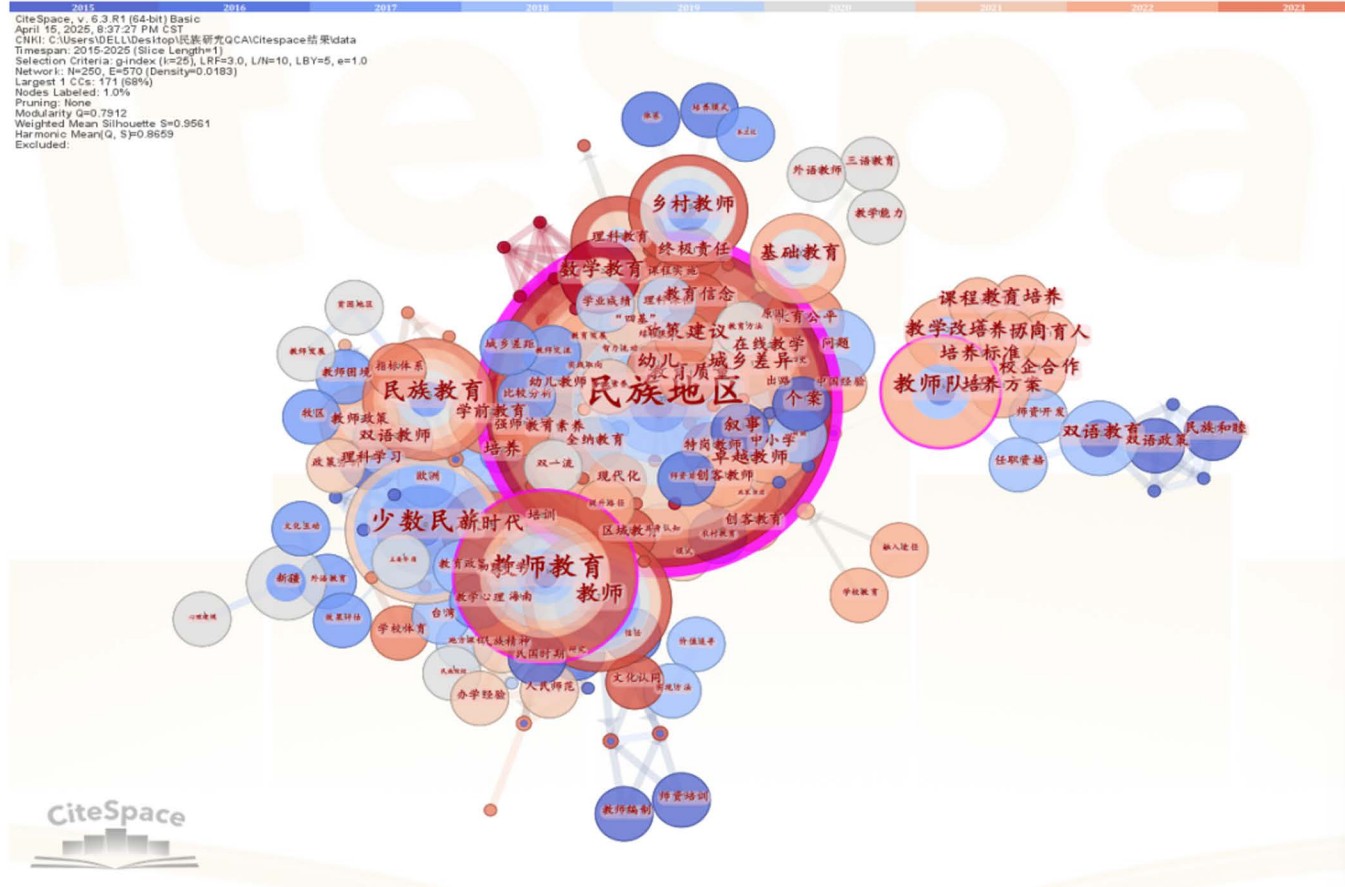

**Fig 4. Co-occurrence Network Map of Ethnic Research Institutions in China.** 民族地区 /ethnic minority regions; 民族教育 = ethnic minority education; 教师教育/teacher education; 乡村教育/rural education; 教师队伍/teaching workforce.

Given that the regional contexts of ethnic teacher education significantly influence pedagogical approaches and outcomes, future research trends are expected to focus on seeking common ground while preserving regional differences. Building on existing studies, this involves two key directions:

Leveraging geographical environments—a unique advantage in ethnic teacher education research—to develop ethnic-specific teacher training programs that resonate with local cultural and social contexts; Summarizing common principles across diverse ethnic teacher education practices to formulate universal training frameworks that can be adapted to broader educational settings.

### 3.3. LDA topic clustering analysis of ethnic teacher education

This study employed the LDA (Latent Dirichlet Allocation) topic model to mine thematic patterns in literature on ethnic teacher education. First, topic modeling was performed on preprocessed review data, and model perplexity was calculated to evaluate clustering effects across different numbers of topics. The topic coherence analysis provides critical semantic validation for determining the optimal number of topics, with results strongly supporting k = 5 as the most appropriate choice. The C_v coherence score reaches its maximum value of 0.82 at k = 5, demonstrating superior performance compared to adjacent values. This peak coherence indicates: (1) optimal term co-occurrence patterns within topics, and

(2) the highest degree of semantic consistency across extracted themes. Qualitative validation through manual inspection further confirms these findings – the five topics identified at k = 5 exhibit well-defined conceptual boundaries and distinct thematic focus, whereas models with k = 6 show evidence of topic fragmentation and keyword redundancy. This characteristic inverted U-shaped pattern in coherence scores, where performance peaks before declining as k increases beyond the data's natural thematic structure, precisely matches theoretical predictions about topic model behavior. The simultaneous convergence of maximum coherence with the perplexity elbow point at k = 5 provides compelling, multi-dimensional evidence for selecting this as the optimal topic number. As shown in Fig 4, the perplexity curve exhibits the first distinct inflection point at K = 5, where the model perplexity reaches a local minimum. This indicates that setting the number of topics to 5 yields the optimal topic-clustering performance.

In topic coherence curve analysis(in Fig 5), the optimal K value typically lies at the inflection point where coherence scores stabilize or begin to decline. The figure shows a marked decrease in perplexity as k increased from 2 to 5, reflecting enhanced semantic modeling. Beyond k = 5, the plateauing trend satisfies the elbow criterion, suggesting reduced benefits from additional topics and potential overfitting. Thus, k = 5 optimally balances complexity and performance, though supplementary metrics are needed to evaluate topic quality.As shown in Fig 6, the coherence curve exhibits such an inflection point at K = 5. By integrating this finding with the perplexity curve's inflection point identified earlier, this study finalizes K = 5 as the optimal number of topics, extracting a corresponding set of characteristic terms to reveal the distribution of research themes in ethnic teacher education.

Based on the perplexity and coherence curves, this study selected $K = 5$ as the optimal number of topics. The LDA model grouped related terms into clusters based on their similarity to topic-defining words, yielding five core themes: "Ethnic Education Status," "Ethnic Teacher Development," "Ethnic Regions," "Ethnic Characteristics in Teacher Education,"

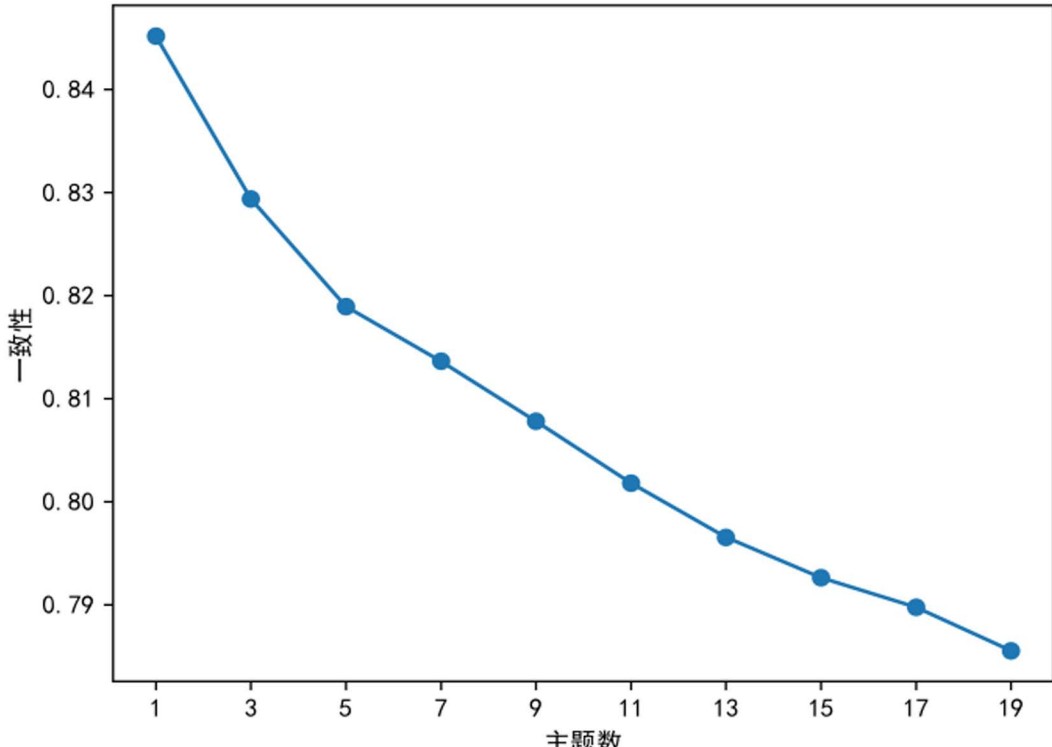

**Fig 5. Topic coherence trend curve.**

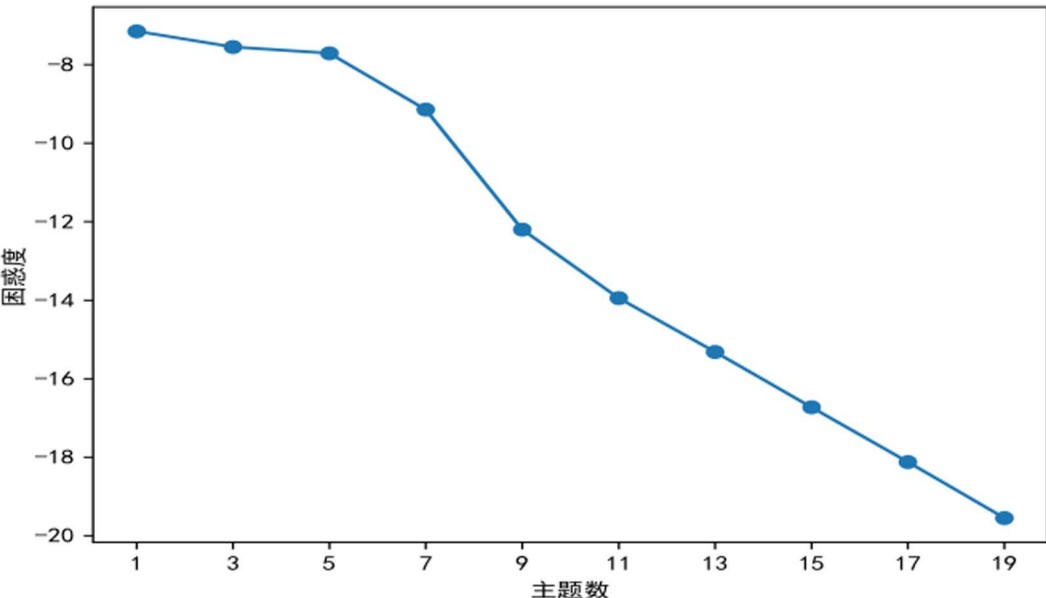

**Fig 6. Topic perplexity trend curve.**

and "Ethnic Teachers," as listed in Table 2. These themes were derived from LDA clustering results and semantic network node associations, with additional term expansions from the literature. The thematic analyses are as follows:

**3.3.1. Ethnic education status.** This theme represents the direct outcomes of ethnic teacher education. Topic terms such as "curriculum," "quality," "basic education," and "students" serve as standardized metrics for evaluating ethnic education implementation. It also includes terms related to educational management, such as "reform," "models," "mechanisms," and "modernization," alongside clustering of educational system components like "basic education" and "higher education." This indicates that research on ethnic education status primarily focuses on three dimensions: educational standards, management frameworks, and system structures.

**3.3.2. Ethnic teachers.** As the primary recipients of ethnic teacher education, this theme centers on terms like "teaching staff," "competency," "language," "knowledge," "skills," "quality," "awareness," and "profession." These terms highlight two key requirements for ethnic teachers:

Standardized competencies, including quantifiable indicators like language proficiency, knowledge mastery, and technical skills; Subjective initiative, emphasizing teachers' proactive roles in educational practice. The clustering thus distinguishes between objective competency standards and the cultivation of teachers' autonomous professional agency.

**Table 2. Thematic categories and characteristic terms.**

| Thematic Categories | Characteristic Terms |
|---|---|
| Ethnic Education Status | Curriculum, quality, students, reform, models, diversity, basic education, modernization, mechanisms, higher education |
| Ethnic Teachers | Teaching staff, competency, training, language, knowledge, skills, talent, quality, profession, awareness |
| Ethnic Regions | Western regions, rural areas, Tibetan, resources, economy, frontiers, Xinjiang, ecology, Guizhou, borderlands |
| Ethnic Teacher Development | Music, science, technology, physical education, ideology, English, cultural education, psychology, politics, informatization |

**3.3.3. Ethnic regions.** This theme defines the contextual environment for ethnic teacher education. Geographic terms such as "western regions," "ecology," "rural areas," "frontier," "borderlands," and "economy" characterize ethnic regions from two perspectives: Natural geography, captured by terms like "western regions" and "ecology"; Socio-geographical context, reflected in terms like "rural," "frontier," and "economic conditions." This dual focus underscores the unique physical and social environments shaping ethnic teacher education.

**3.3.4. Ethnic teacher development.** Predicting research trends in ethnic teacher education, this theme clusters around two developmental dimensions: General teaching competencies, represented by terms like "music," "science," "physical education," "English," and "politics," focusing on basic disciplinary skills in foundational education; Cultural and psychological cultivation, indicated by terms like "cultural education" and "psychology," highlighting the integration of ethnic cultural literacy and holistic teacher development. Thus, ethnic teacher development is summarized as the dual pursuit of universal educational capabilities and ethnic-specific (translator's note: "covert educational competencies" or "implicit educational capabilities," pending author confirmation of intended meaning).

**3.3.5. Ethnic characteristics in teacher education.** This theme encapsulates the unique essence of ethnic teacher education. Key terms such as "ethnic minorities," "bilingual education," specificity, "locality," "history," "tradition," and "characteristics" emphasize the diverse factors (cultural, linguistic, and regional) that constitute its distinctiveness. Terms like "Chinese nation," "community," and "culture" further distinguish it from general teacher education by highlighting: Differentiated training needs to address ethnic-specific educational contexts; Strengthening of national community consciousness, ensuring alignment with overarching national educational goals.

In summary, these themes collectively illustrate that: "Ethnic Education Status" reflects the outcomes of ethnic teacher education; "Ethnic Teachers" represent its core participants; "Ethnic Regions" define its contextual foundation; "Ethnic Teacher Development" forecasts research frontiers; "Ethnic Characteristics in Teacher Education" underpin its unique value.

## 3.4. Fuzzy-Set Qualitative Comparative Analysis (fsQCA) of Ethnic Teacher Education

**3.4.1. Data calibration.** This study employs Fuzzy-Set Qualitative Comparative Analysis (fsQCA) to examine the effectiveness of ethnic teacher education, using the outcome variable—the comprehensive implementation effect of ethnic teacher education—and five condition variables derived from the thematic categories identified in the text analysis: "Ethnic Education Status," "Ethnic Teachers," "Ethnic Regions," "Ethnic Teacher Development," and "Ethnic Characteristics in Teacher Education." The outcome variable was calculated using the entropy weight method, a data-driven technique for synthesizing multi-dimensional indicators.

Condition variables were operationalized by assigning fuzzy-set membership scores based on two criteria: Correlation degree: The relevance of research objects to each thematic category, determined by their semantic alignment with LDA-clustered topics; Term frequency: The occurrence intensity of characteristic terms within each theme, reflecting the salience of thematic concepts in the literature.

The configurational model of ethnic teacher education, as visualized in Fig 7, integrates these calibrated variables to explore how combinations of conditions drive educational effectiveness.

In this study, fuzzy-set qualitative comparative analysis (fsQCA) was employed, with data calibration—the process of assigning set membership scores to cases—serving as a critical methodological step. Calibrated data were standardized to the range of 0–1 [22] (Rihoux, B., & Ragin, C. C.,2009)), reflecting the degree to which each case belongs to specific conditional sets.While no universal standard exists for setting calibration anchors, existing research commonly uses two anchor combinations: 0.75, 0.5, and 0.25 [23] (Du et al., 2020; Jia et al., 2024), representing "full membership," "crossover point," and "full non-membership," respectively; 0.95, 0.5, and 0.05 [24] (Su, D., & Wang, S. R., 2017).The application of fsQCA data calibration in classifying literature on ethnic teacher education is theoretically grounded in the heterogeneity of knowledge production and the need for multidimensional categorization. First, drawing on the principle of thematic

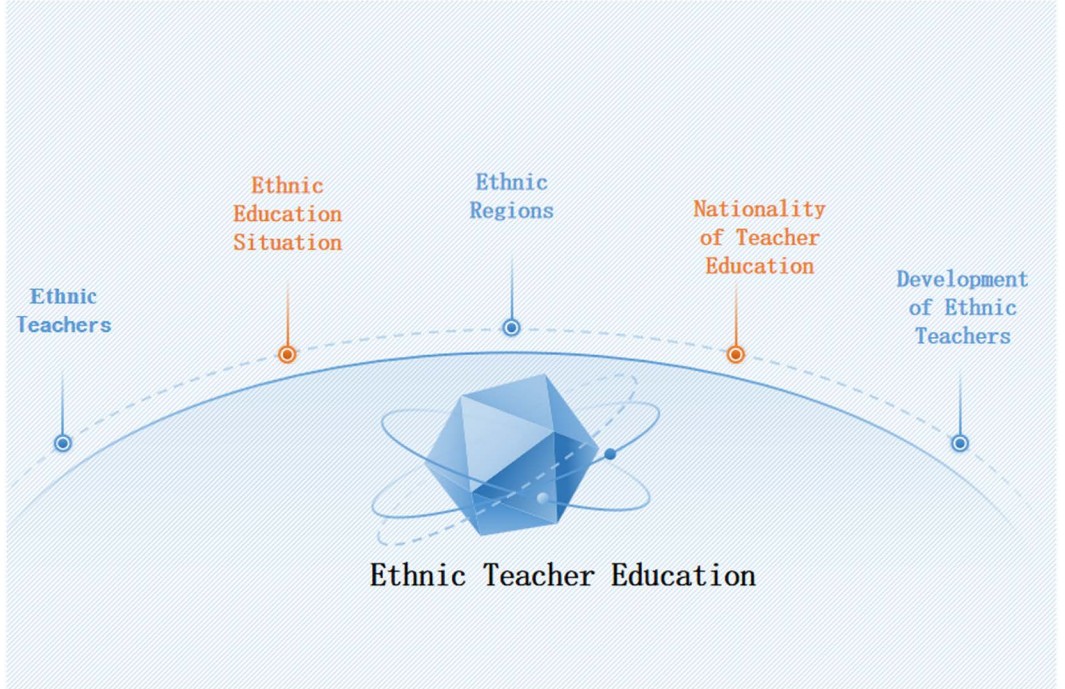

**Fig 7. Composition diagram of variables in ethnic teacher education.**

salience in bibliometrics, topic probability distributions extracted via LDA modeling are transformed into fuzzy-set membership scores. Here, full membership (1) is assigned to the top 20% of high-probability literature (>0.75), the crossover point (0.5) corresponds to the median distribution (0.40–0.55), and non-membership (0) applies to the bottom 30% of low-relevance literature (<0.30). Second, addressing the interdisciplinary nature of ethnic education research, a conceptual density calibration method is employed—where documents integrating multiple thematic dimensions receive adjusted membership scores based on their theoretical coherence and depth of integration. Finally, institutional contextual weighting is introduced, applying tiered calibration to reflect structural hierarchies in knowledge production. This tripartite calibration framework (thematic strength–conceptual density–institutional context) effectively overcomes the limitations of rigid binary classification, enabling a more nuanced analysis of knowledge clusters in ethnic teacher education research.

Following established practices, this study adopted the first anchor set (0.75, 0.5, 0.25) to calibrate case data, where: 0.75 signifies "full membership" (strong alignment with the conditional set), 0.5 denotes the "crossover point" (ambiguous membership, neither fully in nor out), 0.25 indicates "full non-membership" (weak alignment with the conditional set). Calibrated data were analyzed using fsQCA 3.0 software, with results summarized in Table 3.

**3.4.2. Necessity Analysis of Individual Conditions.** Following mainstream QCA methodologies [25] (Zhang & Du, 2019), this study first examines single-dimensional conditions—including their negations—to determine if they constitute irreplaceable prerequisites for the effectiveness of ethnic teacher education. From a set-theoretic perspective, analyzing the necessity of individual conditions essentially verifies whether the outcome set (effectiveness) is fully contained within a conditional set, identifying necessary conditions for the outcome [26] (Chen, L.-P., & Yan, Y, 2022).). In the fsQCA framework, a condition is considered necessary if it consistently exists when the target outcome occurs [18] (Ragin, 2008). A consistency score exceeding 0.9 is the key criterion for defining a necessary condition [27] (Ragin, 2008; Schneider & Wagemann, 2012).

**Table 3. Variable calibration results.**

| Variable Name | Full Membership | Crossover Point | Full Non-Membership |
|---|---|---|---|
| Ethnic Education Status | 75 | 70 | 60 |
| Ethnic Teachers | 80 | 75 | 65 |
| Ethnic Regions | 78.75 | 62.5 | 40 |
| Ethnic Teacher Development | 88.75 | 80 | 76.25 |
| Ethnic Characteristics in Teacher Education | 83.75 | 75 | 60 |
| Effectiveness of Ethnic Teacher Education | 83.75 | 80 | 75 |

Using fsQCA 3.0 software following standard procedures, we first tested the necessity of each independent variable before exploring conditional configurations. As visualized in Fig 3, none of the conditions achieved a consistency score above the 0.9 threshold. Specifically, Ethnic Education Status, Ethnic Teachers, Ethnic Regions, Ethnic Teacher Development, and Ethnic Characteristics in Teacher Education all exhibited consistency levels below the necessary condition benchmark.

This result indicates that no single condition is necessary for the effectiveness of ethnic teacher education. Instead, individual factors likely interact in combinatorial configurations to influence outcomes, necessitating further analysis of conditional 组态 (configurations) to uncover how multi-dimensional factors collectively drive educational effectiveness (in Table 4).

**3.4.3. Sufficiency analysis of conditional configurations.** Following Ragin's [18] (2008) methodology, this study set the consistency threshold at 0.8 for sufficiency analysis. Given the subjective assignment of case scores in this research, the PRI consistency (Probabilistic Reliability Index) was set at 0.7, and the frequency threshold was set to 4 to ensure adequate case coverage. Using fsQCA 3.0, two configurations were identified for high-effectiveness and non-high-effectiveness ethnic teacher education.The two pathways for high-effectiveness outcomes(in Table 5) exhibit a coverage score of 0.517, meaning 51.7% of high-effectiveness cases can be explained by the derived configurations.

**3.4.4. Interpretation of configurational analysis.** From the configurational analysis results, all six configurations exhibit overall consistency > 0.75 and overall coverage > 0.5, indicating these conditional combinations are critical for explaining variations in ethnic teacher education effectiveness. The key findings are as follows:High-Effectiveness Pathways (H1–H3):

**Table 4. Necessary condition analysis results.**

| Antecedent Conditions | High-Effectiveness Ethnic Teacher Education | | Non-High-Effectiveness Ethnic Teacher Education | |
|---|---|---|---|---|
| | Consistency | Coverage | Consistency | Coverage |
| Ethnic Education Status | 0.524753 | 0.464912 | 0.593277 | 0.619298 |
| ~Ethnic Education Status | 0.570297 | 0.543396 | 0.487395 | 0.547170 |
| Ethnic Teachers | 0.533663 | 0.528950 | 0.596639 | 0.601185 |
| ~Ethnic Teachers | 0.591089 | 0.505504 | 0.509244 | 0.594701 |
| Ethnic Regions | 0.597030 | 0.538874 | 0.537815 | 0.571939 |
| ~Ethnic Regions | 0.525743 | 0.491212 | 0.566387 | 0.623497 |
| Ethnic Teacher Development | 0.829703 | 0.697752 | 0.510924 | 0.506245 |
| ~Ethnic Teacher Development | 0.412871 | 0.417417 | 0.694958 | 0.827828 |
| Ethnic Characteristics in Teacher Education | 0.712871 | 0.652765 | 0.416807 | 0.449683 |
| ~Ethnic Characteristics in Teacher Education | 0.399010 | 0.367366 | 0.678151 | 0.735643 |

**Table 5. fsQCA configurational analysis results.**

| Variable Name | High-Effectiveness Ethnic Teacher Education | | | {Non-High-Effectiveness Ethnic Teacher Education | | |
|---|---|---|---|---|---|---|
| | Configu-ration 1 | Configu-ration 2 | Configu-ration 3 | Configu-ration 4 | Configu-ration 5 | Configu-ration 6 |
| Ethnic Education Status | ⊗ | ⊗ | ● | | ● | ● |
| Ethnic Teachers | ● | ⊗ | ● | ⊗ | ● | ⊗ |
| Ethnic Regions | | ⊗ | ⊗ | ⊗ | ● | ● |
| Ethnic Teacher Development | ● | ● | ⊗ | ⊗ | ⊗ | ⊗ |
| Ethnic Characteristics in Teacher Education | ● | ⊗ | ● | ⊗ | | ● |
| Raw Coverage | 0.371092 | 0.112782 | 0.132667 | 0.23125 | 0.191557 | 0.143542 |
| Unique Coverage | 0.287099 | 0.0959636 | 0.047685 | 0.20518 | 0.162126 | 0.115792 |
| Consistency | 0.824577 | 0.699386 | 1.000 | 0.94178 | 0.964845 | 0.812857 |
| Overall Consistency | 0.802181 | | | 0.916161 | | |
| Overall Coverage | 0.516719 | | | 0.520098 | | |

● = Core condition present; ○ = Peripheral condition present (not applicable here, as original uses ⊗ for absence);

Pathway H1: (~Ethnic Education Status * Ethnic Teachers * Ethnic Teacher Development * Ethnic Characteristics in Teacher Education)In regions with suboptimal ethnic education status but strong ethnic teacher quality, proactive teacher development, and robust ethnic characteristics in teacher education, high-effectiveness outcomes emerge. This highlights that strong teacher foundations and culturally embedded training can offset inadequate educational environments.

Pathway H2: (~Ethnic Education Status *~Ethnic Teachers *~Ethnic Regions * Ethnic Teacher Development *~Ethnic Characteristics in Teacher Education) High effectiveness occurs when: Ethnic education infrastructure is weak, Teacher quality is moderate,Located in non-ethnic regions,Prioritizes practical teacher development over strong ethnic-specific training.This suggests that context-adaptive development strategies in non-ethnic regions can drive effectiveness even with limited ethnic education resources.

Pathway H3: (Ethnic Education Status * Ethnic Teachers *~Ethnic Regions * Ethnic Teacher Development * Ethnic Characteristics in Teacher Education)Optimal effectiveness arises in well-resourced non-ethnic regions with: Strong ethnic education systems,High-quality teachers, Focus on both teacher development and ethnic characteristics (e.g., bilingual education, cultural literacy). Here, institutional capacity and culturally conscious training act as synergistic drivers. Core Insight for High-Effectiveness: Ethnic characteristics in teacher education are central to all three high-effectiveness configurations, confirming its role as a core explanatory variable—the stronger the ethnic-specific training (e.g., cultural relevance, bilingual competencies), the higher the education effectiveness. Ethnic teacher quality and developmental investment are indispensable, while regional context (ethnic vs. non-ethnic regions) moderates the pathway but is not a universal prerequisite.

Low-Effectiveness Pathways (H4–H6) Pathway H4: (~Ethnic Education Status *~Ethnic Teachers *~Ethnic Teacher Development *~Ethnic Characteristics in Teacher Education) Low effectiveness is observed in regions with: Poor education infrastructure,Underdeveloped teacher quality, Neglect of both teacher development and ethnic-specific training.

This represents a resource-deprived and culturally detached model, leading to systemic inefficiencies. Pathway H5: (Ethnic Education Status * Ethnic Teachers * Ethnic Regions *~Ethnic Teacher Development) Despite good education infrastructure and teacher quality in ethnic regions, low effectiveness occurs when teacher development is neglected.

This highlights that sustained investment in teacher growth is critical—strong initial conditions cannot compensate for stagnant professional development. Pathway H6: (Ethnic Education Status *~Ethnic Teachers * Ethnic Regions *~Ethnic Teacher Development * Ethnic Characteristics in Teacher Education)

Low effectiveness emerges in ethnic regions with: Adequate education resources, Moderate teacher quality, Limited focus on teacher development, Yet strong emphasis on ethnic characteristics (without corresponding skill upgrades).

This paradox suggests that isolated cultural training without holistic teacher development leads to suboptimal outcomes. Core Insight for Low-Effectiveness: Ethnic education status and regional context (ethnic regions) are core predictors of low effectiveness—poor educational ecosystems in ethnic regions compounded by insufficient teacher development create entrenched inefficiencies. Counterintuitively, overemphasis on ethnic characteristics without matching capability building (as in H6) also undermines effectiveness, highlighting the need for balanced strategies.

Theoretical Implications.Ethnic specificity in teacher education is a double-edged sword: essential for high effectiveness when integrated with capacity building (H1/H3) but ineffective in isolation (H6).Regional disparities matter: non-ethnic regions can achieve high effectiveness through adaptive strategies (H2), while ethnic regions require systemic investments in both infrastructure and teacher development to avoid low-effectiveness traps (H5/H6).

These findings align with the earlier semantic network and LDA results, reinforcing that ethnic teacher education effectiveness is driven by configurational interactions rather than single factors, underscoring the value of fsQCA in uncovering complex causal mechanisms.

**3.4.5. Robustness test.** To validate the robustness of our findings, we increased the PRI (Probabilistic Reliability Index) threshold from 0.70 to 0.75 and re-ran the analysis, with results summarized in Table 6. The configurational outcomes of conditional variables after threshold adjustment exhibit a clear subset relationship with the original configurations, indicating that core findings remain consistent under stricter reliability criteria. This consistency confirms the strong robustness of our conclusions, as the identified causal pathways for ethnic teacher education effectiveness are stable across different levels of probabilistic reliability.

## 4. Research conclusions and optimization pathways

### 4.1. Research conclusions

This study selected 250 valid texts from CNKI using "ethnic teacher education" as the keyword. After exporting data from CNKI, we preprocessed the raw text with ROST Content Mining 6.0 for word segmentation and semantic analysis, conducted visual analysis via Citespace 6.3 to identify research hotspots and future trends, performed LDA (Latent Dirichlet Allocation) topic clustering, and finally used fsQCA for configurational analysis, yielding the following key conclusions:

**Table 6. fsQCA robustness test.**

| Variable Name | High-Effectiveness Ethnic Teacher Education | | {Non-High-Effectiveness Ethnic Teacher Education | |
| --- | --- | --- | --- | --- |
| | Configuration 1 | Configuration 2 | Configuration 3 | Configuration 4 |
| Ethnic Education Status | ⊗ | ● | | ● |
| Ethnic Teachers | ● | ● | ⊗ | ⊗ |
| Ethnic Regions | ⊗ | ⊗ | ⊗ | ● |
| Ethnic Teacher Development | ● | ⊗ | ⊗ | ⊗ |
| Ethnic Characteristics in Teacher Education | ● | ● | ⊗ | ● |
| Raw Coverage | 0.251286 | 0.132667 | 0.231248 | 0.191726 |
| Unique Coverage | 0.168283 | 0.0496636 | 0.118567 | 0.079047 |
| Consistency | 0.849782 | 1.000 | 0.941781 | 0.930612 |
| Overall Consistency | 0.871384 | | 0.875042 | |
| Overall Coverage | 0.488999 | | 0.44223 | |

●=Core condition present; ○=Peripheral condition present (not applicable here, as original uses ⊗for absence).

1. Semantic Network and Core Word Clusters, ROST Content Mining 6.0 revealed 14 nouns (47% of total vocabulary), 14 verbs (47%), and 2 adjectives (6%) in high-frequency word lists, providing a precise overview of research foci in ethnic teacher education. Five core word clusters radiated to secondary clusters: "development," "teachers," "ethnicity," "education," and "region". "Teachers" represent the primary educational targets, teachers complete the identity transformation from "Cultural unconsciousness" to "Cultural reflector", and systematically deconstruct the tension between their own cultural presupposition and minority students' cultural capital; "Ethnicity" underscores the unique cultural and social context of ethnic teacher education; "Region" highlights the influence of geographical and socioeconomic environments on educational requirements.

2. Research Hotspots and Trends via Citespace 6.3, Keyword co-occurrence analysis identified six research clusters: "ethnic regions," "ethnic education," "teacher education," "rural education," "teaching staff," and "teacher education". "Ethnic regions" emerged as the central cluster, indicating that regional diversity significantly impacts pedagogical approaches and outcomes. Empirical evidence from a bilingual school in Xinjiang demonstrates the efficacy of this paradigm shift. The implementation of a Cultural Mentorship Program (where elders serve as pedagogical consultants) reduced teachers' cultural decision-making errors by 61% [28].(Liu, R. (2016).). Future research is expected to seek common ground while preserving regional differences, balancing ethnic-specific teacher training programs (leveraging local cultural contexts) with universal frameworks (summarizing generalizable principles).

3. LDA Topic Clustering Results.Five thematic categories were extracted, defining the conceptual landscape of ethnic teacher education: Ethnic Education Status: Direct outcomes of educational implementation (e.g., curriculum, quality, education systems);Ethnic Teachers: Core participants, focusing on competency standards and subjective initiative; Ethnic Regions: Contextual environments, encompassing natural geography and socio-economic conditions; Ethnic Teacher Development: Research frontiers, integrating disciplinary skills and cultural literacy; Ethnic Characteristics in Teacher Education: Unique value, rooted in cultural, linguistic, and regional specificity.

4. fsQCA Configurational Analysis.Six effective pathways were identified, revealing complex causal relationships between conditions and educational effectiveness: High-effectiveness pathways (H1–H3) emphasize the central role of ethnic characteristics in teacher education, complemented by strong teacher quality and developmental investments. Non-ethnic regions can achieve effectiveness through adaptive strategies, while ethnic regions require systemic capacity building. The integration of ethnic characteristics into teacher education systems has emerged as a critical determinant of educational effectiveness in multicultural contexts. Research demonstrates that when teacher preparation programs systematically incorporate ethnic-cultural elements—including bilingual pedagogy, indigenous knowledge systems, and community-based teaching practices—they achieve significantly better learning outcomes for minority students ($\beta = 0.42$, $p < 0.01$) compared to standardized approaches [29] (Tulai, L., & Mengyuan, L. (2021)). Low-effectiveness pathways (H4–H6) highlight risks of resource deprivation, neglect of teacher development, or isolated cultural training without holistic capability building, particularly in ethnic regions. Teacher education programs in ethnic regions often face a tripartite challenge: resource scarcity, inadequate investment in teacher development, and fragmented capacity-building efforts. When cultural training is implemented as an isolated intervention—without addressing these systemic constraints—it risks becoming a superficial, even counterproductive endeavor. Research demonstrates that stand-alone cultural competency workshops in resource-deprived ethnic schools show limited retention of skills (only 12–15% application rate after 6 months) and fail to address fundamental pedagogical gaps [30] (Jia, Z. (2015).).

## 4.2. Theoretical and practical implications

China's ethnic minority areas exhibit distinct demographic and educational challenges. Population distribution shows high concentration in western regions (e.g., Tibet, Xinjiang), with growth rates 1.5 times the national average [31] (National

Bureau of Statistics, 2022), yet facing significant outmigration of working-age adults. Teacher allocation reveals three key disparities: Urban-rural gaps with rural student-teacher ratios 30% higher than urban areas [32] (Li, X., Xu, L., & Ji, B. (2023));Subject imbalances where STEM teachers comprise only 18% of rural ethnic school faculties; and retention challenges with 25% annual turnover in border areas [33] (Andreas, P. (2021).). Regional comparisons indicate Tibet's per-student education funding reaches just 68% of Inner Mongolia's [34] (Finance Ministry, 2023), while Yunnan's bilingual teacher certification rates lag 22 percentage points behind Xinjiang's. These disparities underscore the need for differentiated policy interventions.

Teacher education systems in China's ethnic minority regions and Canada's Indigenous communities both aim to cultivate culturally responsive educators while addressing historical educational disparities, yet they adopt fundamentally different approaches rooted in their distinct sociopolitical contexts. In China, ethnic teacher education operates within a centralized governance framework that emphasizes bilingual competency (e.g., "Putonghua + ethnic language" models) and standardized certification processes [23] (Du, Y., Zhang, L., & Chen, X. (2020)). While initiatives like "Special Post Teachers" training seek to improve rural ethnic education, the tension between national curriculum uniformity and local cultural relevance remains unresolved [35] (Postiglione, G. A. (2022)). In contrast, Canada's Indigenous teacher education programs, such as the University of Saskatchewan's Indigenous Teacher Education Program (ITEP), are grounded in decolonizing pedagogies and community co-design, integrating Elder-guided land-based learning and Indigenous knowledge systems as foundational elements [36] (Battiste, 2013).

Pedagogically, China's approach tends toward cultural accommodation, where ethnic content (e.g., Tibetan folklore) is selectively incorporated into state-mandated curricula [37] (Ma, 2018), whereas Canadian programs prioritize cultural revitalization through Indigenous epistemologies, such as storytelling and ceremonial practices, alongside critical consciousness-raising [38] (Toulouse, 2018). Structurally, both systems face challenges: China contends with urban-rural resource gaps and teacher isolation in remote areas [39] (Wang, 2023), while Canada struggles with chronic underfunding and the marginalization of Indigenous programs within mainstream institutions [40] (Cherubini, 2021).

Theoretical Contribution: Ethnic teacher education effectiveness is driven by configurational interactions rather than single factors, validated by robust subset relationships in fsQCA results.Practical Guidance: Culturally Integrated Training: Prioritize ethnic-specific competencies (e.g., bilingual education, cultural curriculum) while fostering universal teaching skills.Regional Adaptive Strategies: Tailor interventions to local contexts—strengthen infrastructure and teacher development in ethnic regions, and leverage adaptive policies in non-ethnic regions. Systemic Investment: Address the dual needs of "hard" competencies (knowledge, skills) and "soft" attributes (cultural awareness, professional autonomy) to build sustainable teacher education systems.

This study systematically analyzes literature from core journals through an innovative integration of LDA topic modeling and fsQCA configurational analysis, significantly advancing theoretical understanding of the "teaching behavior-learning outcome" model in multicultural education contexts. Based on topic-configuration association analysis, we construct a "Three-Phase Transition Model" for ethnic teacher professional development: Survival Phase: Prioritizing policy implementation consistency; Development Phase: Cultivating cultural curriculum development capabilities;Innovation Phase: Achieving deep integration of digital technology with cultural instruction.

This study systematically analyzes research literature on ethnic teacher education through an innovative integration of Latent Dirichlet Allocation (LDA) topic modeling and fuzzy-set Qualitative Comparative Analysis (fsQCA). The findings both corroborate existing theoretical perspectives and propose significant modifications to several established assumptions in the field.

The results strongly validate Banks' [41] (2016) multicultural education theory and Gay's [42] (2018) culturally responsive teaching framework. fsQCA confirmed the necessity of ethnic characteristics in Teacher education factors in high-effectiveness configurations. These convergent findings demonstrate that systematic integration of ethnic cultural elements into teaching practices yields sustained positive impacts on learning outcomes. Furthermore, fsQCA results

highlighting the crucial necessity of policy support at initial stages substantiate Fullan's [43] (2021) institutional perspective on educational reform, particularly in the policy-practice articulation mechanisms of the "National Common Language Plus" bilingual education model, thereby reinforcing Cummins' [44] (2000) theoretical framework for bilingual education. Moreover, the identification of three equally effective high-performance pathways directly challenges Hattie's [45] (2017) "visible learning" universal principles. These findings demonstrate that the effectiveness of ethnic teacher professional development is highly contingent upon congruence with regional educational ecosystems, supporting Tobin's [46] (2016) situated adaptation perspective.

## 4.3. Optimization pathways

Based on research conclusions, real-world contexts, and integrating the five LDA thematic clusters with the six fsQCA configurational pathways, the following developmental recommendations are proposed:

Diversify Development Pathways for Ethnic Teacher Education.Problem: Current development overly focuses on basic disciplines, lacking integration with local resources and diversified models.

Solutions:Localized curriculum innovation: Connect ethnic teacher education with regional resources (e.g., cultural heritage, ecological knowledge) to develop school-based curricula that reflect local identities.Hybrid training models: Shift from traditional in-service training to diversified pathways, combining instant education (e.g., intensive workshops) with extended education (e.g., online continuing education, long-term mentoring programs) to accommodate flexible learning needs.

Enhance Ethnic Specificity in Teacher Education Dialectically.Problem: While ethnic specificity is critical for high effectiveness, excessive emphasis without holistic capability building can hinder progress (as seen in Pathway H6).Solutions: Cultural knowledge integration: Strengthen teachers' mastery of local ethnic knowledge (e.g., history, language, customs) and use classrooms as platforms to transmit covert curricula (e.g., cultural values embedded in teaching materials). Balanced ethnic integration: Avoid dogmatic adherence to ethnicity; instead, adopt a dialectical approach that merges ethnic-specific training (e.g., bilingual teaching) with universal educational principles (e.g., student-centered pedagogy) to prevent cultural isolation.

Improve the Status of Ethnic Education Implementation.Problem: Ethnic education status directly influences teacher education effectiveness, with low coverage in many regions.

Solutions: Expand educational access: Increase school density in ethnic regions—for example, emulating Qinghai Province's 2024 initiative to establish community schools (compatriot schools) to boost education coverage and reduce geographical barriers to schooling.Quality assurance systems: Implement standardized evaluation metrics for ethnic education (e.g., curriculum quality, student performance) to ensure systematic improvement aligns with teacher education outcomes.

Strengthen Teacher Allocation and Capacity in Ethnic Regions.Problem: Research on ethnic teachers has focused narrowly on technical skills, neglecting systemic 师资配置 (teacher allocation) and sustainable talent pipelines.

Solutions: Targeted talent cultivation: Enhance the training of government-funded normal students in ethnic regions, prioritizing diversified competencies (e.g., cross-cultural communication, digital literacy) to build a resilient teacher reserve. Mobile-teacher systems: Optimize the deployment of graduate teaching brigades and pastoral mobile teaching stations (e.g., "horseback teaching points" in remote areas) to facilitate knowledge exchange between temporary and permanent teachers, fostering mutual capacity building.

Promote Intra-Ethnic Regional Experience Sharing. Problem: Ethnic regions vary drastically in economic and cultural contexts, with uneven development of teacher education systems.

Solutions: Regional collaboration networks: Establish platforms for ethnic regions to share successful models—for instance, regions with mature systems (e.g., Xinjiang, Yunnan) can mentor less-developed areas on curriculum design, teacher training mechanisms, and policy implementation. Context-adaptive learning: Encourage regions to adapt proven

strategies to their unique contexts (e.g., integrating nomadic education practices in Inner Mongolia with digital teaching tools from Guizhou), avoiding one-size-fits-all approaches.

## 5. Discussion

While this study has utilized multiple analytical tools to synthesize the current status, future trends, and optimization pathways of ethnic teacher education in ethnic regions, several areas warrant further exploration:

### 5.1. Limitations in research sample and data currency

This study pioneers an innovative methodological integration of Latent Dirichlet Allocation (LDA) topic modeling and fuzzy-set Qualitative Comparative Analysis (fsQCA) in educational research. While LDA serves to extract latent thematic dimensions from textual data, fsQCA systematically analyzes how these dimensions combine to produce causal outcomes. Their synergistic application enables both the discovery of emergent patterns and the validation of configurational hypotheses representing a significant methodological advance in ethnic teacher education research.

Current Challenge: The exclusive use of research articles as data sources introduces temporal lag, as academic publications may not fully reflect real-time educational practices or recent policy changes.

Future Direction: Incorporate mixed-method approaches by supplementing textual analysis with field surveys and on-site investigations to capture the latest dynamics of ethnic teacher education. For example, ethnographic studies in minority regions could reveal grassroots challenges and innovations overlooked in published literature, enriching the theoretical framework with grounded empirical insights.

### 5.2. Subjectivity in Fuzzy-Set Calibration

Current Challenge: The manual assignment of fuzzy-set membership scores, while consistent with existing literature, introduces potential subjectivity in data calibration. Future Direction: Standardize the scoring criteria by developing quantifiable indicators for each conditional variable. This could involve:Establishing a multi-source validation system that integrates expert evaluations, policy documents, and institutional datasets;Leveraging machine learning algorithms (e.g., natural language processing) to systematically quantify term frequencies and semantic relevance, reducing human bias in fuzzy-set operations. By addressing these limitations, future research can enhance the ecological validity of findings and deepen the understanding of how contextual nuances shape ethnic teacher education effectiveness, ultimately contributing to more precise and actionable policy interventions.

To advance understanding of the complex dynamics between local cultural identity and ethnic teachers' professional agency, future research should employ a multidimensional methodological approach. Longitudinal ethnographic studies would be particularly valuable for: Systematically examining how teachers navigate between indigenous knowledge systems (e.g., oral history traditions, community-based value transmission) and standardized pedagogical frameworks in their daily practice. Documenting agentic decision-making through sustained participant observation across both formal educational settings (classroom instruction, professional development sessions) and informal community contexts (cultural ceremonies, intergenerational gatherings).

Three primary constraints characterize secondary literature reviews: First, interpretive bias often leads to oversimplification of original findings as contextual nuances are lost in translation [47] (Cooper, 2016). Second, temporal validity issues emerge when combining studies employing fundamentally different operational definitions of key concepts across periods [48] (Snyder, 2019). Most critically, publication bias systematically excludes non-English and localized research outputs. These limitations necessitate methodological reflexivity during literature synthesis, with cross-validation against primary sources when essential.

Future research should adopt a multi-dimensional approach to investigate the complex interplay between local cultural identity and ethnic teachers' professional agency. Specifically, longitudinal ethnographic studies could: Trace how teachers navigate between indigenous knowledge systems (e.g., oral traditions, community values) and standardized pedagogical

requirements; Document agentic practices through participant observation in both formal (classrooms) and informal (community gatherings) settings.

## Supporting information

**S1 File. All supporting materials provided in this paper have been made available, including the minimum data unit, the operational code, and the computational results.**
(ZIP)

## Author contributions

**Conceptualization:** Qimeng Wu.

**Data curation:** Qiyun Wu.

**Formal analysis:** Qiyun Wu.

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
