## [Decision Letter · Decision Letter 0]

23 Jun 2025

Dear Dr. Wu,

Thank you for submitting your manuscript to PLOS ONE. After careful consideration, we feel that it has merit but does not fully meet PLOS ONE’s publication criteria as it currently stands. Therefore, we invite you to submit a revised version of the manuscript that addresses the points raised during the review process.

We look forward to receiving your revised manuscript.

Kind regards,

Muhammad Zammad Aslam, Ph.D.

Academic Editor

PLOS ONE

Additional Editor Comments (if provided):

Reviewers' comments:

Reviewer's Responses to Questions

**Comments to the Author**

1. Is the manuscript technically sound, and do the data support the conclusions?

Reviewer #1: Yes

Reviewer #2: Yes

2. Has the statistical analysis been performed appropriately and rigorously?

Reviewer #1: Yes

Reviewer #2: Yes

3. Have the authors made all data underlying the findings in their manuscript fully available?

Reviewer #1: Yes

Reviewer #2: Yes

4. Is the manuscript presented in an intelligible fashion and written in standard English?

Reviewer #1: Yes

Reviewer #2: Yes

Reviewer #1: I. ABSTRACT ANALYSIS

Abstract was already Systematic and complete. Has a background, specific research objectives, and appropriate methodology

Abstract Improvement Suggestions:

1. Add data sources and sampling techniques concisely: “...based on a sample of 250 peer-reviewed articles retrieved from the CNKI database…”

2. Emphasize the purpose in the opening sentence: Add one sentence stating the gap in previous research and the unique contribution of this study. Example: “Previous studies have rarely examined ethnic teacher education from a configurational perspective. This study addresses that gap…”

3. Emphasize the unique novelty this research at the end of abstract

4. Clarify theoretical and practical contributions more explicitly: Use sentences such as: “The findings contribute to both theory and policy by proposing a novel framework for…”

5. Language can be condensed to be more focused: Some sentences in the middle of the abstract are rather long and can be made more concise without reducing the meaning.

II. INTRODUCTION ANALYSIS

Strengths:

1. Systematic (Explanation starts from the importance of ethnic education → role of teachers → government policy → challenges in the field).

2. Logical writing (The sequence flows logically: from policy → challenges → research gaps → new approaches).

3. Relevant and up-to-date topics (Bringing up policies from 2018 and 2022. This is relevant and current in the Chinese context).

4. There is a clear research gap: It is stated that previous studies: are too retrospective, only assess current practices, do not address future needs, and are not systemic.

5. References: Some references are up-to-date (2018–2022), but some are too old or general (e.g. Cremin's theory from the 1970s). Some also come from local journals or Chinese sources, not international journals.

Weaknesses

1. The formulation of the problem is not explicit.

The article explains the challenges and objectives, but does not formulate the research question in an explicit sentence such as: "This study aims to answer the question: how does the configuration of variables... affect the effectiveness of..."

2. Lack of reinforcement from international literature.

The explanation uses a lot of data and studies from China. It would be stronger if there was a comparison with similar international contexts (e.g., teacher education in indigenous communities, Africa, Latin America).

Suggestions for Improvement

1. Add an explicit statement of the problem statement: For example: “Given the fragmented understanding of ethnic teacher education in China, this study addresses the following research questions: (1) What are the core thematic dimensions of ethnic teacher education? (2) How do combinations of these dimensions affect the effectiveness of ethnic teacher education?”

2. Strengthen the international literature base: Add 1–2 references from international contexts (e.g., from the International Journal of Educational Development or Comparative Education) to demonstrate global relevance.

3. Rationalize the approach more explicitly: Explain why the combination of LDA + fsQCA is more appropriate than classical quantitative approaches (e.g., regression), especially for the complexity of ethnic regions.

III. METHODOLOGY ANALYSIS

Strength

1. Innovative approach and fit with the research question: fsQCA is very appropriate for looking at combinations of variables in complex contexts such as ethnic teacher education. LDA helps to find hidden themes in a data-driven way, without researcher bias.

2. Utilization of big data: Using 250 articles over 10 years provides a strong basis for identifying trends.

3. Validation of results is done through robust tests (increasing the PRI threshold in fsQCA).

Weaknesses

1. There is no validation process for the quality of articles used as data sources. It is not stated whether the articles went through blind review, categorization of articles from locally indexed journals, or came from, Scopus, etc.

2. Subjectivity in fuzzy-set calibration. Although using a reference (0.75 / 0.5 / 0.25), the mapping of values from articles to the scale is still manual and can be biased.

3. Does not include triangulation or inter-coder validation. There is no inter-rater reliability check if there are several people reading and assessing the article.

4. No mention of software testing or training (ROST/CiteSpace/LDA) for model validation or parameter settings.

Suggestions for Methodology Improvement

1. Add article inclusion-exclusion criteria: For example: “Only peer-reviewed journal articles related to ethnic teacher training in mainland China were included…”

2. Use supporting instruments such as checklists: For example: coding tools to assess the relevance, methodological quality, and focus of the selected articles.

3. Explain the data selection validation process: Who selected? Was double-blind screening done? Were there other reviewers?

4. Add references to the quality of the LDA and fsQCA methods: So that it can be understood why the K5 parameters were selected and how the fsQCA was calibrated.

5. Expand data triangulation: Add, for example, interviews with ethnic teacher lecturers or policy documents, so that the findings are stronger and not only based on academic literature.

IV. DISCUSSION ANALYSIS

Strengths and weaknesses

1. The discussion is quite complete: 1) Explains the results of fsQCA in detail, including: Six configurations (3 high effective, 3 low effective), Determinants and their relationships with each other; 2) Accompanied by interpretation of each configuration, both success and failure conditions; 3) Theoretical and practical implications are provided.

However, there are some incomplete ones: • Does not discuss differences between ethnic regions specifically (eg Tibet vs Xinjiang vs Guizhou); • Does not explore the possible role of educational actors such as principals, communities, or NGOs;

2. Has strong statistical data: • Uses data from 250 scientific articles filtered from CNKI; • LDA analysis produces 5 dominant topics based on the perplexity and coherence models; • fsQCA uses fuzzy-set scoring based on word frequency and correlation; • Coverage, consistency values, and a very complete table of result configurations are available.

However: • There is no statistical data in quantitative field form (e.g. teacher survey results, questionnaires, or hypothesis tests). • All analyses are based on secondary literature — this makes the data technically strong, but less empirically direct.

3. Some results are linked to appropriate references: • The discussion is quite deep in terms of internal interpretation of the data. • However, it is rare to explicitly link the findings to previous studies, especially: Not comparing whether the results are similar or different from other studies. There are no citations explaining the position of this article among similar articles (both national and international).

4. The references are sufficient in number, but not optimal in terms of supporting the results. • There are around 15 references, mostly local literature (China). • There are almost no international references or meta-analyses of ethnic education. • Not used to strengthen the results, more as a general background or methodology.

Suggestions for Improvement of Discussion

1. Strengthen connections with relevant literature, especially: 1) Add comparisons with studies from other countries (eg: teacher education for indigenous communities in Canada, indigenous tribes in Australia, etc.); 2) Use previous study citations to support or contrast the results of the fsQCA configuration.

2. Include theoretical references when explaining the configuration of results, For example, when mentioning the importance of teacher development and cultural education - connect it with the theory of culturally responsive pedagogy (Gay, 2010) or teacher agency.

3. Provide additional contextual data, such as: 1) Demographic conditions of ethnic areas, 2) Challenges of teacher distribution, 3) Training gaps between regions - so that readers get a real picture.

4. Add citations from national reports or surveys (if any) that describe the situation of ethnic teachers in the field.

5. Provide an explicit statement about the limitations of generalizing the results from secondary literature.

V. CONCLUSION ANALYSIS

Although the RQ is not explicitly written, but the conclusion answers clearly: thematic dimensions are important + how the combination of factors works in shaping educational effectiveness. Novelty is seen through the combination method (LDA + fsQCA) and the configuration of the results. However, it does not explicitly state the scientific contribution that distinguishes this study from others. Already mentioning the shortcomings of the study (1) data only from secondary literature, (2) fsQCA calibration is subjective. Also provides suggestions for further research (using mixed methods: field surveys, automated NLP to reduce human bias, ethnographic studies in the field). The writing is solid, organized, and concise - making it easier for readers to understand the direction of the research and its applications

Suggestions for Improving the Conclusion

1. Add an explicit sentence about novelty:

2. State the theoretical contribution explicitly: For example: expanding the understanding of the teaching behavior-learning outcome model in the context of multicultural education.

3. Emphasize the position of the results in the academic landscape: Do these results support or challenge previous findings?

4. Provide a follow-up research question: Example: "Future research should explore how local cultural identity directly affects the professional agency of ethnic teachers using field-based ethnographic methods."

Reviewer #2: The manuscript presents a technically robust mixed-methods investigation into ethnic teacher education in China. The authors combine computational text analysis (ROST Content Mining, CiteSpace) with LDA topic modeling and fuzzy-set Qualitative Comparative Analysis (fsQCA). The methodology is well-justified and appropriate for the research aims. The sample of 250 articles, drawn from a larger CNKI dataset, is systematically processed and filtered, and the data analysis clearly aligns with the research questions. The conclusions—namely, the identification of five thematic clusters and six optimization pathways—are logically derived from the data and are supported by both quantitative and qualitative evidence. However, the authors should further clarify the rationale for the sample size and provide more details on the operationalization of “Ethnic Characteristics in Teacher Education” during fsQCA calibration.

The statistical analysis is rigorous and transparent. The fsQCA is conducted according to established standards, with clear reporting of calibration anchors (0.75, 0.5, 0.25), consistency thresholds, and coverage scores. The necessity and sufficiency analyses are appropriately executed, and robustness checks (e.g., adjusting the PRI threshold) confirm the stability of the findings. The manuscript would benefit from including a supplementary table with raw data (e.g., topic-term distributions) and a discussion of potential subjectivity in manual fuzzy-set scoring, possibly referencing intercoder reliability if available.

The authors state that all relevant data are included in the manuscript and its Supporting Information files. This satisfies PLOS ONE’s data policy, provided that the processed datasets (such as high-frequency word lists and LDA matrices) are indeed included. If the original CNKI texts cannot be shared due to third-party restrictions, this should be explicitly stated in the Data Availability Statement, as required by PLOS ONE policy.

The manuscript is generally clear and written in standard English, but minor revisions are needed for grammar and style. Typographical errors (e.g., “Doctoer” instead of “Doctor”) should be corrected, complex sentences should be simplified, and references to figures and tables should be checked for consistency and completeness. Professional language editing is recommended to further improve clarity and readability.

**Do you want your identity to be public for this peer review?** For information about this choice, including consent withdrawal, please see our Privacy Policy

Reviewer #1: No

Reviewer #2: No

---

## [Author Response · Author response to Decision Letter 1]

1 Jul 2025

1.ABSTRACT

1.Add data sources and sampling techniques concisely: “...based on a sample of 250 peer-reviewed articles retrieved from the CNKI database…”

A: We have added the following to the abstract: 'This study analyzes 250 peer-reviewed articles from PKU/CSSCI-indexed journals in the CNKI database (2015–2025), identified using keywords such as "ethnic teacher education," "ethnic education," and "ethnic teachers." These were refined from an initial pool of 2,672 records through manual screenin.

2.Emphasize the purpose in the opening sentence: Add one sentence stating the gap in previous research and the unique contribution of this study. Example: “Previous studies have rarely examined ethnic teacher education from a configurational perspective. This study addresses that gap…”

A: We have added the following to the abstract: “Previous studies have rarely examined ethnic teacher education from a configurational perspective. This study addresses that gap.”

3.Emphasize the unique novelty this research at the end of abstract

A: We have added the following to the abstract: “This study introduces the novel integration of LDA topic modeling and fsQCA in educational research. While LDA uncovers latent themes and fsQCA examines causal complexity, their combined application enables simultaneous discovery and validation of configurations a previously unexplored approach in ethnic teacher education.”

4.Clarify theoretical and practical contributions more explicitly: Use sentences such as: “The findings contribute to both theory and policy by proposing a novel framework for…”

A: We have added the following to the abstract: “These findings make dual contributions: theoretically, by advancing a novel conceptual framework; and practically, by yielding actionable policy implications for ethnic teacher education development.”

5.Language can be condensed to be more focused: Some sentences in the middle of the abstract are rather long and can be made more concise without reducing the meaning.

A: We have restructured the abstract, converting lengthy single sentences into concise, digestible statements.

1.INTRODUCTION

1.Add an explicit statement of the problem statement: For example: “Given the fragmented understanding of ethnic teacher education in China, this study addresses the following research questions: (1) What are the core thematic dimensions of ethnic teacher education? (2) How do combinations of these dimensions affect the effectiveness of ethnic teacher education?”

A: We have added the following to the introduction “Given the fragmented understanding of ethnic teacher education in China, this study examines: (1) the core thematic dimensions of ethnic teacher education, and (2) how their configurations influence educational effectiveness.

2.Strengthen the international literature base: Add 1–2 references from international contexts (e.g., from the International Journal of Educational Development or Comparative Education) to demonstrate global relevance.

A: The references section has been reorganized and expanded to include more internationally contextualized sources.

3.Rationalize the approach more explicitly: Explain why the combination of LDA + fsQCA is more appropriate than classical quantitative approaches (e.g., regression), especially for the complexity of ethnic regions.

A: We have added the following to the introduction “ This study innovatively integrates LDA and fsQCA to overcome regression analysis' limitations in studying ethnic teacher education. Unlike regression's linear assumptions, our approach captures system complexity and equifinality through two phases: LDA extracts cultural themes from qualitative data, then fsQCA analyzes their configurations. Key advantages include: (1) small-N robustness with contextual sensitivity , (2) identification of causal asymmetries, and (3) multiple success pathways for diverse contexts. This advances understanding of ethnic education's complex causality beyond variable-centered methods.”

3. METHODOLOGY

1. Add article inclusion-exclusion criteria: For example: “Only peer-reviewed journal art6b icles related to ethnic teacher training in mainland China were included…”

A: The exclusion criteria for article selection have been elaborated with greater specificity.“This study exclusively analyzed peer-reviewed journal articles focusing on teacher training programs within Mainland China. ”

2.Use supporting instruments such as checklists: For example: coding tools to assess the relevance, methodological quality, and focus of the selected articles.

A: We employed a simple scale tool as a reference during the screening process.We have added the following to the methodology “ Article selection followed PRISMA guidelines for duplicate removal, with two researchers independently screening studies using the CASP Qualitative Checklist. Discrepancies were resolved through consensus.”

4.Explain the data selection validation process: Who selected? Was double-blind screening done? Were there other reviewers?

A: We have added the following to the methodology “ Article selection followed PRISMA guidelines for duplicate removal, with two researchers independently screening studies using the CASP Qualitative Checklist. Discrepancies were resolved through consensus.”

5.Add references to the quality of the LDA and fsQCA methods: So that it can be understood why the K5 parameters were selected and how the fsQCA was calibrated.

A: We have provided more detailed citations and explanations regarding the perplexity and consistency curves on pages 11-12 of the main text.

6.Expand data triangulation: Add, for example, interviews with ethnic teacher lecturers or policy documents, so that the findings are stronger and not only based on academic literature.

A: We sincerely appreciate your insightful suggestions and fully acknowledge their validity. However, due to current limitations in our geographic location, research funding, and personnel availability, we are unable to conduct interviews at this stage. We have therefore explicitly addressed this constraint in the Discussion section and identified it as an important direction for future research.

4. DISCUSSION

1. Strengthen connections with relevant literature, especially: 1) Add comparisons with studies from other countries (eg: teacher education for indigenous communities in Canada, indigenous tribes in Australia, etc.); 2) Use previous study citations to support or contrast the results of the fsQCA configuration.

A: We have incorporated the following additions to strengthen the international and methodological dimensions of our study: Page 25: Added a case study on Indigenous education in Canada;Page 26: Included comparative analysis with fsQCA (Fuzzy-Set Qualitative Comparative Analysis) computational results

2. Include theoretical references when explaining the configuration of results, For example, when mentioning the importance of teacher development and cultural education - connect it with the theory of culturally responsive pedagogy (Gay, 2010) or teacher agency.

A: We have conducted a linkage analysis of culturally responsive pedagogy on page 22 of the main text, with supporting citations (Citation 29).

3. Provide additional contextual data, such as: 1) Demographic conditions of ethnic areas, 2) Challenges of teacher distribution, 3) Training gaps between regions - so that readers get a real picture.

A: We have included an analysis of educational realities and research disparities in China's ethnic regions on page 22 of the manuscript.

4. Add citations from national reports or surveys (if any) that describe the situation of ethnic teachers in the field.

A: We sincerely appreciate your constructive suggestions. After carefully reviewing numerous national reports and academic publications, we were unable to identify suitable references that directly align with the proposed modifications. While we fully acknowledge the value of your recommendations, current limitations in available literature prevent their immediate implementation. We will prioritize this as a key objective for our ongoing research program.

5.Provide an explicit statement about the limitations of generalizing the results from secondary literature.

A: We have addressed the limitations of secondary literature research on page 26 of the manuscript and outlined plans to enhance data diversity in future studies.

5.CONCLUSION

1. Add an explicit sentence about novelty:

A: We have articulated the study's novel contributions in a dedicated section on page 25 of the manuscript.This study pioneers an innovative methodological integration of Latent Dirichlet Allocation (LDA) topic modeling and fuzzy-set Qualitative Comparative Analysis (fsQCA) in educational research. While LDA serves to extract latent thematic dimensions from textual data, fsQCA systematically analyzes how these dimensions combine to produce causal outcomes. Their synergistic application enables both the discovery of emergent patterns and the validation of configurational hypotheses representing a significant methodological advance in ethnic teacher education research.

2. State the theoretical contribution explicitly: For example: expanding the understanding of the teaching behavior-learning outcome model in the context of multicultural education.

A: We have elaborated on the study's theoretical contributions across pages 25-26 of the manuscript.

3. Emphasize the position of the results in the academic landscape: Do these results support or challenge previous findings?

A: We have addressed the study's academic positioning through a comparative analysis on page 23 of the manuscript.

4. Provide a follow-up research question: Example: "Future research should explore how local cultural identity directly affects the professional agency of ethnic teachers using field-based ethnographic methods."

A:We have dedicated the final section of the manuscript to outlining future research directions, with your valuable suggestions prioritized as key investigative pathways. Future research should adopt a multi-dimensional approach to investigate the complex interplay between local cultural identity and ethnic teachers' professional agency. Specifically, longitudinal ethnographic studies could: Trace how teachers navigate between indigenous knowledge systems (e.g., oral traditions, community values) and standardized pedagogical requirements; Document agentic practices through participant observation in both formal (classrooms) and informal (community gatherings) settings.

---

## [Decision Letter · Decision Letter 1]

13 Jul 2025

Contrasting and Prioritizing Dimensions in Ethnic Teacher Education: A Convergent Analysis with LDA and fsQCA

PONE-D-25-25749R1

Dear Dr. Wu,

We’re pleased to inform you that your manuscript has been judged scientifically suitable for publication and will be formally accepted for publication once it meets all outstanding technical requirements.

Kind regards,

Muhammad Zammad Aslam, Ph.D.

Academic Editor

PLOS ONE

Additional Editor Comments (optional):

Reviewers' comments:

Reviewer's Responses to Questions

**Comments to the Author**

Reviewer #1: (No Response)

Reviewer #2: All comments have been addressed

2. Is the manuscript technically sound, and do the data support the conclusions?

Reviewer #1: Yes

Reviewer #2: No

3. Has the statistical analysis been performed appropriately and rigorously?

Reviewer #1: Yes

Reviewer #2: Yes

4. Have the authors made all data underlying the findings in their manuscript fully available?

Reviewer #1: Yes

Reviewer #2: Yes

5. Is the manuscript presented in an intelligible fashion and written in standard English?

Reviewer #1: Yes

Reviewer #2: Yes

Reviewer #1: Revision review:

I. Abstract Revision: Done

II. Introduction Revision: Done

III. Methodology Revision: Done

IV. Discussion Revision : Done

V. Conclusion Revision: Done

VI. References: done, from 15 to 50

Reviewer #2: I appreciate how you have taken my comments and suggestions into account in the article, and I can clearly see all the improvements that have been implemented.

**Do you want your identity to be public for this peer review?** For information about this choice, including consent withdrawal, please see our Privacy Policy

Reviewer #1: No

Reviewer #2: No

---

## [Editor Report · Acceptance letter]

PONE-D-25-25749R1

PLOS ONE

Dear Dr. Wu,

I'm pleased to inform you that your manuscript has been deemed suitable for publication in PLOS ONE. Congratulations! Your manuscript is now being handed over to our production team.

Kind regards,

on behalf of

Dr. Muhammad Zammad Aslam

Academic Editor

PLOS ONE